# Aligning cellular and molecular components in age-dependent tertiary lymphoid tissues of kidney and liver

**Naoya Toriu**[1,2], **Yuki Sato**[1,3¤], **Hiroteru Kamimura**[4], **Takahisa Yoshikawa**[1], **Masaou Tanaka**[1], **Shinya Yamamoto**[1], **Shingo Fukuma**[5,6], **Masakazu Hattori**[7], **Shuji Terai** [4], **Motoko Yanagita** [1,2]*

**1** Department of Nephrology, Graduate School of Medicine, Kyoto University, Kyoto, Japan, **2** Institute for the Advanced Study of Human Biology (ASHBi), Kyoto University, Kyoto, Japan, **3** Medical Innovation Center TMK Project, Graduate School of Medicine, Kyoto University, Kyoto, Japan, **4** Division of Gastroenterology and Hepatology, Graduate School of Medical and Dental Sciences, Niigata University, Niigata, Japan, **5** Human Health Sciences, Graduate School of Medicine, Kyoto University, Kyoto, Japan, **6** Department of Epidemiology Infectious Disease Control and Prevention, Hiroshima University Graduate school of Biomedical and Health Sciences, Higashihiroshima, Japan, **7** Department of Immunosenescence, Graduate School of Medicine, Kyoto University, Kyoto, Japan

¤ Current address: Division of Immunology and Rheumatology, Mayo Clinic, Rochester, Minnesota, United States of America
* motoy@kuhp.kyoto-u.ac.jp

## Abstract

Tertiary lymphoid tissues (TLTs) are ectopic lymphoid structures induced by multiple stimuli, including infection and tissue injuries; however, their clinical relevance in disease progression has remained unclear. We demonstrated previously that TLTs develop in mouse and human kidneys with aging and can be a potential marker of kidney injury and prognosis, and therapeutic targets. In addition, we found that two types of unique lymphocytes that emerge with aging, senescence-associated T cells and age-associated B cells, are essential for TLT formation in the kidney. Although TLTs develop with aging in other organs as well, their cellular and molecular components, and clinical significance remain unclear. In the present study, we found that TLTs developed in the liver with aging, and that their cellular and molecular components were similar to those in the kidneys. Notably, senescence-associated T cells and age-associated B cells were also present in hepatic TLTs. Furthermore, analysis of publicly available data on human liver biopsy transcriptomes revealed that the expression of TLT-related genes was elevated in the liver biopsy samples from hepatitis C virus (HCV)-infected patients compared with those without HCV infection and was associated with liver injury and fibrosis. Therefore, we analyzed liver biopsy samples from 47 HCV patients and found that TLTs were present in 87.2% of cases and that the numbers and stages of TLTs were higher in aged patients and cellular and molecular components of TLTs in humans were similar to those in mice. Our findings suggesting that age-dependent TLT formation is a systemic phenomenon across the tissues and aging is also a predisposing factor for TLT formation across organs.

**Data availability statement:** The mouse data-sets are available in the Supporting Information files (S2_Table). The human datasets that support the findings of this study are available from the corresponding author, but restrictions apply to the availability of these data, which were used under license for the current study, and so are not publicly available. Data are however available from the authors upon reasonable request and with permission of the ethics committee at Niigata University and the ethical committee at Kyoto University Hospital. The institutional contacts are: ethics@adm.niigata-u.ac.jp (the ethics committee at Niigata University) and ethcom@kuhp.kyoto-u.ac.jp (the ethical committee at Kyoto University Hospital). Previously published RNA-seq data are available in GEO (GSE84346: https://www.ncbi.nlm.nih.gov/geo/query/acc.cgi?acc=GSE84346 and GSE171294: https://www.ncbi.nlm.nih.gov/geo/query/acc.cgi?acc=GSE171294).

**Funding:** This research was supported by the Japan Agency for Medical Research and Development (AMED) under Grant Number AMED-CREST 19gm0610011, 22gm1210009, 22zf0127003h001, 22lm0203006, and 22ek0310020h0001; KAKENHI Grant-in-Aids for Scientific Research B (20H03697). This work was partly supported by the World Premier International Research Center Initiative (WPI), MEXT, Japan. The funders had no role in study design, data collection and analysis, decision to publish, or preparation of the manuscript.

**Competing interests:** YS was employed by the TMK project, a collaboration project between Kyoto University and Mitsubishi Tanabe Pharma. MY has received research grants from Mitsubishi Tanabe Pharma and Boehringer Ingelheim. MH was employed by the Immunosenescence project, which was a collaboration project between Kyoto University and Ono Pharmaceutical Co. Ltd. Other authors report no conflicts of interest.

## Introduction

Tertiary lymphoid tissues (TLTs) are ectopic lymphoid structures induced in non-lymphoid organs under chronic inflammatory conditions such as cancer, autoimmunity, and infection [1–6]. TLTs have similar structural and functional properties with lymph nodes [7], mainly composed of lymphocytes, which are functionally and structurally supported by non-hematopoietic stromal cells such as fibroblasts [8,9]. TLTs have the potential to initiate adaptive immune responses against locally presented antigens and modify tissue injury. Their roles are context dependent: TLTs induced by infections are considered to mount protective immune responses and be beneficial for the host. However, the failure to eradicate pathogens can lead to the development of autoimmunity and be maladaptive [1].

In previous studies, we demonstrated that TLTs develop with aging, and the interaction between two unique age-dependent lymphocytes, senescence-associated T (SAT) cells [10] and age-associated B cells (ABCs) [11], plays a critical role in their formation [12]. These cells accumulate within TLTs and communicate with each other via CD153–CD30 signaling, which is essential for TLT formation. We also found that TLT formation proceeds through at least three developmental stages, and are associated with the severity of kidney injury [13]. We also demonstrated that TLTs are frequently found in kidney transplants, and that advanced-stage TLTs are associated with poor graft outcomes [14]. While TLTs development is also reported in various other tissues, such as the liver, lung, thyroid gland, and pancreas [9,15–21], their clinical relevance remains unclear.

The presence of lymphocyte aggregates around the portal vein in hepatitis C virus (HCV) hepatitis has been reported since the late 1980s [22]. They have been referred to as lymphoid follicles, lymphoid aggregates, or portal tract inflammation. Lymphoid follicles/aggregates are frequently observed in liver biopsy samples from patients with HCV hepatitis and even more frequent in active hepatitis [23–28]. Some reports have further proved the activation of T and B cells [17] and the clonal expansion of B cells in lymphoid follicles/aggregates in HCV hepatitis [18,29,30]. However, the correlation between lymphoid follicles/aggregates and liver damage and prognosis has not been sufficiently proven. In addition, the definition of a lymphoid follicle/aggregate varies from report to report and includes a wide range from simple inflammatory cell infiltration to those including a germinal center.

In this study, we tried to characterize the cellular and molecular components of hepatic TLTs using superaged mouse and human livers, and to elucidate the clinical relevance of them. We first analyzed TLTs in superaged mouse livers. We confirmed the similarities of their cellular and molecular components, including SAT cells and ABCs, the above-mentioned major drivers of TLT formation, with those of TLTs in aged kidneys. Next, we analyzed publicly available data on human liver biopsy transcriptomes. We found that the expression of TLT-related genes was elevated in liver biopsies from HCV patients compared with those in non-infected patients. Based on this finding, we further analyzed liver biopsy samples from patients with HCV hepatitis based on our definition and staging method of TLTs [13]. We found that TLTs were formed in 87.2% of liver biopsies and associated with older age. Further we found that the cellular and molecular components of liver TLTs in humans were also similar with those in mice.

## Materials and methods

### Animals

To represent young and superaged mice, we utilized 2-month-old and 23–24-month-old C57BL6J male mice as well as 2-month-old and 16–23-month-old *Spp1*-EGFP-KI mice, respectively. We purchased C57BL6J mice from Japan SLC. *Spp1*-EGFP-KI mice have been

described previously [10]. All mice were maintained under specific pathogen-free conditions in the animal facility at Kyoto University. The mice were euthanized under anesthesia with an intraperitoneal injection of a mixture of midazolam, butorphanol tartrate, and medetomidine hydrochloride. All animal experiments were approved by the Animal Research Committee, Graduate School of Medicine, Kyoto University (MedKyo18528, MedKyo19205, MedKyo20187, MedKyo21189, MedKyo22182, MedKyo22589, MedKyo23153), and were conducted in accordance with the United States National Institutes for Health Guide for the Care and Use of Laboratory Animals. The study was carried out in compliance with the ARRIVE guidelines.

## Histochemistry

Left lateral lobes of the mouse livers were harvested, fixed in 10% neutral-buffered formalin (NBF), embedded in paraffin, sectioned (3.0-μm thickness) using the microtome (RM2235; Leica, Nussloch, Germany), and stained with hematoxylin and eosin (H&E) [31]. Human livers were fixed in 10% NBF, embedded in paraffin, sectioned (3.0-μm thickness) using the microtome (RM2125RT; Leica), and stained with H&E and silver stain [31].

## Immunofluorescence

For immunofluorescence studies of mouse livers, the livers were fixed in 4% paraformaldehyde, incubated in 20% sucrose for 6 h, and further incubated in 30% sucrose in PBS at 4°C overnight. OCT-embedded (Sakura Finetek, Tokyo, Japan) livers were cryosectioned into 6.0-μm sections using the cryostat (CM3050 S; Leica) and mounted on Superfrost slides (Matsunami Glass, Osaka, Japan). Sections were blocked with 5% serum appropriate for the secondary antibody for 1 h at room temperature, and then were incubated overnight at 4°C with primary antibodies as follows: anti-LYVE-1 (catalog no. ab14917; Abcam, Cambridge, UK), -α-SMA (catalog no. C6198; Sigma-Aldrich, Missouri, USA), -CD3ε (catalog no. 550275; BD PharMingen, San Jose, CA, USA), -B220 (catalog no. 557390; BD PharMingen), -CD21 (catalog no. ab75985; Abcam), -Ki67 (catalog no. 14-5698; eBioscience, San Diego, CA, USA, catalog no. ab15580; Abcam), -CD45 (catalog no. 14-0451; eBioscience), -p75NTR (catalog no. AF1157; R&D Systems, Minneapolis, MN, USA), -Desmin (catalog no. PA5-16705; Thermo Fisher Scientific, Waltham, MA, USA), -CXCL13 (catalog no. AF470; R&D Systems), -CCL19 (catalog no. AF880; R&D Systems), -CD11c (catalog no. 550283; BD PharMingen), -p21 (catalog no. 188224; Abcam), -CD90.2 (catalog no. 553011; BD PharMingen), and -cytokeratin 19 (catalog no. ab52625; Abcam) antibodies. In addition, EGFP was visualized with the anti-GFP antibody (catalog no. ab13970; Abcam). Staining was visualized using the appropriate secondary antibodies and samples counterstained with DAPI.

For immunofluorescence studies of human livers, liver samples were fixed in 10% NBF and embedded in paraffin. Paraffin-embedded sections were deparaffinized with xylene, rehydrated, and then steam-heated for 15 min. These sections were incubated with 5% serum appropriate for the secondary antibody for 1 h at room temperature and then were incubated at 4°C with primary antibodies as follows: anti-CD3ε(catalog no. ab5690; Abcam), -CD20 (catalog no. 14-0202; eBioscience), -Ki67 (catalog no. ab16667; Abcam), -CD23 (catalog no. CD23-1B12-L-CE-H, Leica), -Desmin (catalog no. PA5-16705; Thermo Fisher Scientific), –α–SMA (catalog no. C6198; Sigma-Aldrich), -CXCL13 (catalog no. AF801; R&D Systems), and -CD45 (catalog no. 14-9457; eBioscience) antibodies. Staining was visualized using the appropriate secondary antibodies and samples were counterstained with DAPI.

All immunofluorescence samples were analyzed using a confocal microscope (FV1000D; Olympus, Tokyo, Japan: LSM900; ZEISS, Dresden, Germany).

## Immunohistochemistry

For immunohistochemistry, endogenous peroxidase was blocked using 3% $H_2O_2$. Tissue sections were stained with anti-CD21 antibody (catalog no. ab75985; Abcam). Antibody labeling was detected using a diaminobenzidine substrate kit (Vector Laboratories, Newark, CA, USA). The sections were counterstained with hematoxylin. Immunohistochemistry samples were analyzed under an inverted microscope (BZ-X710; Keyence, Osaka, Japan)

## RNA in situ hybridization (RNA-ISH)

Mouse *Tnfsf8*, *Tnfrsf8*, *Cd3e*, and *Cd19* were detected using paraffin-embedded sections of mouse livers and human *TNFSF8* was detected using paraffin-embedded sections of human livers utilizing RNAscope Multiplex Fluorescent Assay V2 (Advanced Cell Diagnostics, Newark, CA, USA) and RNA scope Target probes Mm-Tnfsf8 (498721; Advanced Cell Diagnostics), Mm-CD30 (529811; Advanced Cell Diagnostics), Mm-Cd3e-C2 (314721-C2; Advanced Cell Diagnostics), Mm-Cd19-C3 (314711-C3; Advanced Cell Diagnostics) and Hs-TNFSF8 (439661; Advanced Cell Diagnostics) according to the manufacturer's instructions. RNA-ISH samples were analyzed under a confocal microscope (LSM900; ZEISS, Dresden, Germany).

## Identification and quantification of TLTs

In the present study, we defined TLTs as organized lymphocyte aggregates with signs of proliferation. Quantification of TLT numbers and stages was performed according to a previous publication in a blinded manner [13]. Briefly, we examined immunofluorescence in two serial sections of (1) CD3ε and B220 and (2) Ki67 and CD21 in mice, or (1) CD3ε and CD20 and (2) Ki67 and CD23 in humans [32]. TLT stage determination was performed based on the presence of CD21$^+$ or CD23$^+$ follicular dendritic cells (FDCs) and germinal centers (Ki67$^+$ cell clusters in B-cell areas within TLTs). TLTs containing neither FDCs nor germinal centers were defined as stage I, whereas TLTs that contained FDCs but lacked germinal centers were defined as stage II. TLTs with prominent FDCs and germinal centers were defined as stage III. In mice, the numbers of TLTs are shown as the numbers per unit area of the sample (per mm$^2$), and the relative TLT size is shown by dividing the total TLT size by the sample size. In humans, the relative numbers of TLTs are shown by dividing the TLT numbers by the portal tract numbers, which were counted in the H&E-stained serial section, as described previously [29]. The TLT size and the sample size were measured using Adobe Photoshop software (Adobe, San Jose, CA, USA).

## Real-time PCR

RNA extraction and real-time PCR were performed as described previously [33,34]. The primer sequences are listed in S1 Table. Expression levels were normalized to those of *Gapdh* and expressed relative to the levels in young mouse liver.

## Analysis of human liver specimens

In total, 50 patients with HCV infection who underwent liver biopsy prior to interferon therapy at Niigata University Medical and Dental Hospital between May 1, 2007, and October 31, 2013, were examined. Minors were not included among the patients. Informed consent was obtained using an opt-out process on the website. Clinical data were accessed from February 24 to 26 in 2019 for research purposes. The authors had access to information that cloud identify participants during data collection. After data collection, the patient information was anonymized, and the authors did not have access to information that could identify individual participants. Patients who were also infected with hepatitis B virus infection (n = 1) or those who received immunosuppressive

therapies (n = 2) were excluded. The final study population included 47 patients. We defined young patients as those less than 60 years of age and aged patients as those 60 years of age or more, as described previously [13]. All liver specimens were evaluated using the new Inuyama classification, which classifies chronic hepatitis as the activity grade from A0 to A3 and the fibrosis stage from F0 to F4, as assessed by an experienced hepatic pathologist in a blinded manner [35]. Sustained virological response at 24 weeks (SVR24) was defined as negative for serum HCV RNA at 24 weeks after the end of interferon treatment. Albumin–bilirubin (ALBI) score was calculated using the following formula: $(\log_{10}$ bilirubin $[\mu mol/L] \times 0.66) + ($albumin $[g/L] \times -0.085)$, while ALBI grade was defined by the resulting score ($\leq -2.60 =$ grade 1;$> -2.60$ to $\leq -1.39 =$ grade 2;$> -1.39 =$ grade 3) [36]. The fibrosis-4 (FIB-4) index was calculated using the following formula: age (years) $\times$ glutamic oxaloacetic transaminase $[IU/l]/($platelets $[10^9/l] \times ($glutamic pyruvic transaminase $[IU/l])^{1/2})$ [37]. All human specimens were procured and analyzed after informed consent and with the approval of the ethics committee at Niigata University (2017-0313). Data analysis was also approved by the ethical committee at Kyoto University Hospital (R0254). The study was performed in accordance with the Declaration of Helsinki.

## Reanalysis of published human RNA-seq data

Normalized RNA-seq data available from the National Center for Biotechnology Information's Gene Expression Omnibus (accession number GSE84346 and GSE171294) were further analyzed.

Of the 25 patients with HCV infection who had undergone liver biopsy, there were pre-interferon (pre-IFN) samples for 22 patients. We omitted one case in which the annotation table did not match the ID. We reanalyzed 21 patients with HCV infection and six patients without HCV infection who had undergone a diagnostic liver biopsy. Among the six patients without HCV infection, four underwent a liver biopsy because of elevated liver enzymes and two underwent a liver biopsy because of liver metastases. Furthermore, we reanalyzed 4 patients with hepatitis B virus (HBV)-associated liver cancer and 4 patients with hemangioma as control. All of them had underwent surgical liver resection. The detailed method of data normalization has been described in previously published articles [38,39].

## Statistical analysis

Data are presented in the tables and figures as the number (percentage) for categorical variables and the median and interquartile range (IQR) for continuous variables. Categorical variables were compared using Fisher's exact test. Continuous variables were compared using the Mann–Whitney test or the Kruskal-Wallis test, as appropriate. Correlations were determined using Pearson's correlation analysis. Kaplan-Meier method and log-rank test were used to depict and compare the cumulative incidence of hepatocellular carcinoma (HCC). Statistical analyses were performed using JMP PRO software (version 15.2.0; SAS Inc., Cary, NC, USA) and GraphPad Prism for Mac (version 9.50; GraphPad Software, La Jolla, CA, USA). For all analyses, $p$-values less than 0.05 were considered statistically significant. The datasets of mice used and analyzed in the current study are uploaded as S2 Table.

## Results

### Cellular and molecular components of TLTs in aged livers are similar to those in aged kidneys

To investigate whether aging-associated TLTs formation is also observed in the liver, we first investigated the livers of young (2-month-old) and superaged (23 to 24-month-old) mice.

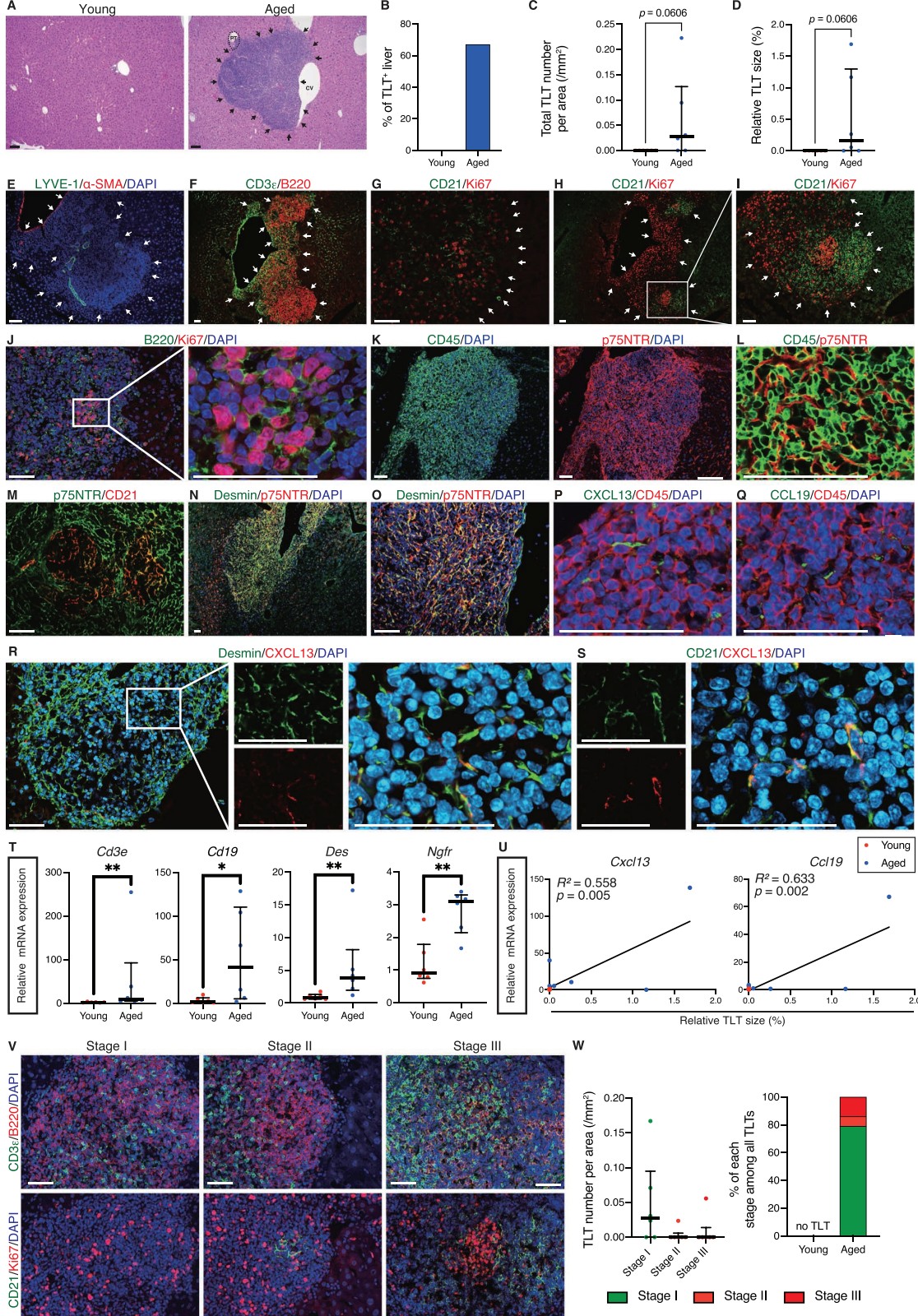

**Fig 1. Superaged mouse develops hepatic TLTs without injury.** (A) Hematoxylin and eosin (H&E) staining. Quantitative analysis of (B) TLT frequencies, (C) TLT numbers, and (D) relative TLT sizes in young and superaged mouse livers (n = 6 per group).

Immunofluorescence analysis of (E) lymphatic vessel endothelial hyaluronan receptor 1 (LYVE-1) and smooth muscle actin (α-SMA); (F) CD3ε (a T-cell marker) and B220 (a B-cell marker); (G–I) CD21 (a follicular dendritic cell (FDC) marker) and Ki67; (J) B220 and Ki67; (K, L) CD45 and p75 neurotrophin receptor (p75NTR); (M) p75NTR and CD21; (N, O) Desmin (a hepatic stellate cell marker) and p75NTR; (P) CXCL13 (a homeostatic chemokine) and CD45; (Q) CCL19 (a homeostatic chemokine) and CD45; (R) Desmin and CXCL13; and (S) CD21 and CXCL13. (T) mRNA expression of *Cd3e*, *Cd19*, *Des* (*Desmin*), and *Ngfr* (*p75ntr*) in young and superaged mouse livers (n = 6 per group). (U) Correlations between relative TLT sizes and mRNA levels of *Cxcl13* and *Ccl19* in young and superaged mouse livers (n = 6 per group). The mRNA expression levels are normalized to those of *Gapdh* and expressed relative to those of the young mouse. (V) Representative immunofluorescence of three distinct stages (I, II, and III) of TLTs. Immunofluorescence analysis of CD3εand B220; CD21 and Ki67 in each stage are serial sections. Stage I TLTs are the aggregation of lymphocytes with the sign of proliferation, and do not include FDCs or germinal centers; Stage II TLTs include FDCs, but not germinal centers; Stage III TLTs include FDCs and germinal centers. (W) The relative numbers of stages I, II, and III TLTs and the proportions of stages I, II, and III TLTs in superaged mouse livers without injury (n = 6). In (C, D, T, W) data are shown as the median and interquartile range. Mann–Whitney test (C, D, T) was used to analyze the difference. Correlations in (U) are determined by Pearson's correlation analysis. TLT number is shown as the number per area of section. *$p \leq 0.05$ and **$p \leq 0.01$. Arrows indicate TLT localization. Scale bars: 50 μm. Abbreviations: CV, central vein; PT, *portal tract.

While young mice did not exhibit TLTs in the liver, 67% of superaged mice exhibited multiple hepatic TLTs, located near portal tracts and central veins (Fig 1A–D). Immunofluorescence analysis of superaged mouse livers showed that hepatic TLTs contained arteries and lymphatic vessels (Fig 1E). Hepatic TLTs were mainly composed of B cells and T cells (Fig 1F), and some TLTs contained CD21+ FDCs (Fig 1G). Moreover, germinal centers, proliferative B-cell clusters, were also detected in some TLTs (Fig 1H–J). Fibroblasts inside the TLTs expressed p75 neurotrophin receptor (p75NTR) (Fig 1K, L), but some TLTs had p75NTR negative area, which was compensated by CD21+ FDCs (Fig 1M), as observed in renal TLTs [9]. A finding characteristic of hepatic TLTs was that Desmin, a marker of hepatic stellate cells (HSCs), was strongly expressed in most p75NTR+ fibroblasts (Fig 1N, O), while CD90.2, a marker of portal fibroblasts, was not well expressed (S1 Fig). The expression of homeostatic chemokines, CXCL13 and CCL19, was confined within TLTs and these were expressed by non-hematopoietic cells, as in the kidney (Fig 1P, Q). CXCL13 was expressed in Desmin+ fibroblasts and CD21+ FDCs (Fig 1R, S). In addition, the expression of *Cd3e*, *Cd19*, *Des* (*Desmin*), and *Ngfr* (*P75ntr*) was upregulated in superaged livers (Fig 1T), and the expression of *Cxcl13* and *Ccl19* was correlated moderately with TLT size in superaged livers (Fig 1U). To further characterize TLTs, we classified TLTs into three phenotypically distinct stages based on the presence or absence of FDCs and germinal centers, as recently reported for renal TLTs [13]. We found stage I, II, and III TLTs in superaged mouse livers (Fig 1V), and demonstrated that 79% of hepatic TLTs in superaged mice were stage I TLTs (Fig 1W). These results suggest that cellular and molecular components of hepatic TLTs were similar to those of renal TLTs. Hepatic TLTs exhibited various stages of development and maturation but, as in the kidney [13], most aging-associated TLTs were stage I.

## Senescence-associated T cells and age-associated B cells were present within hepatic TLTs

As discussed in the Introduction, we previously reported that SAT cells and ABCs accumulated within TLTs in aged injured kidneys [12], and played essential roles in TLT expansion. We, therefore, investigated whether SAT cells were also present in hepatic TLTs using superaged EGFP-*Spp1*-knockin (KI) reporter mice [10]. In superaged EGFP-*Spp1*-KI reporter mice, CD45+GFP+ cells, considered as SAT cells [10], were detectable almost exclusively within the TLTs in the liver and in the kidney (Fig 2A–C). CD45+GFP+ cells were also detectable in superaged spleen, as previously reported (Fig 2D) [10]. In TLTs in the superaged liver, GFP+ cells were colocalized with CD3ε, but not with other cell lineage markers (Fig 2E–J). Additionally,

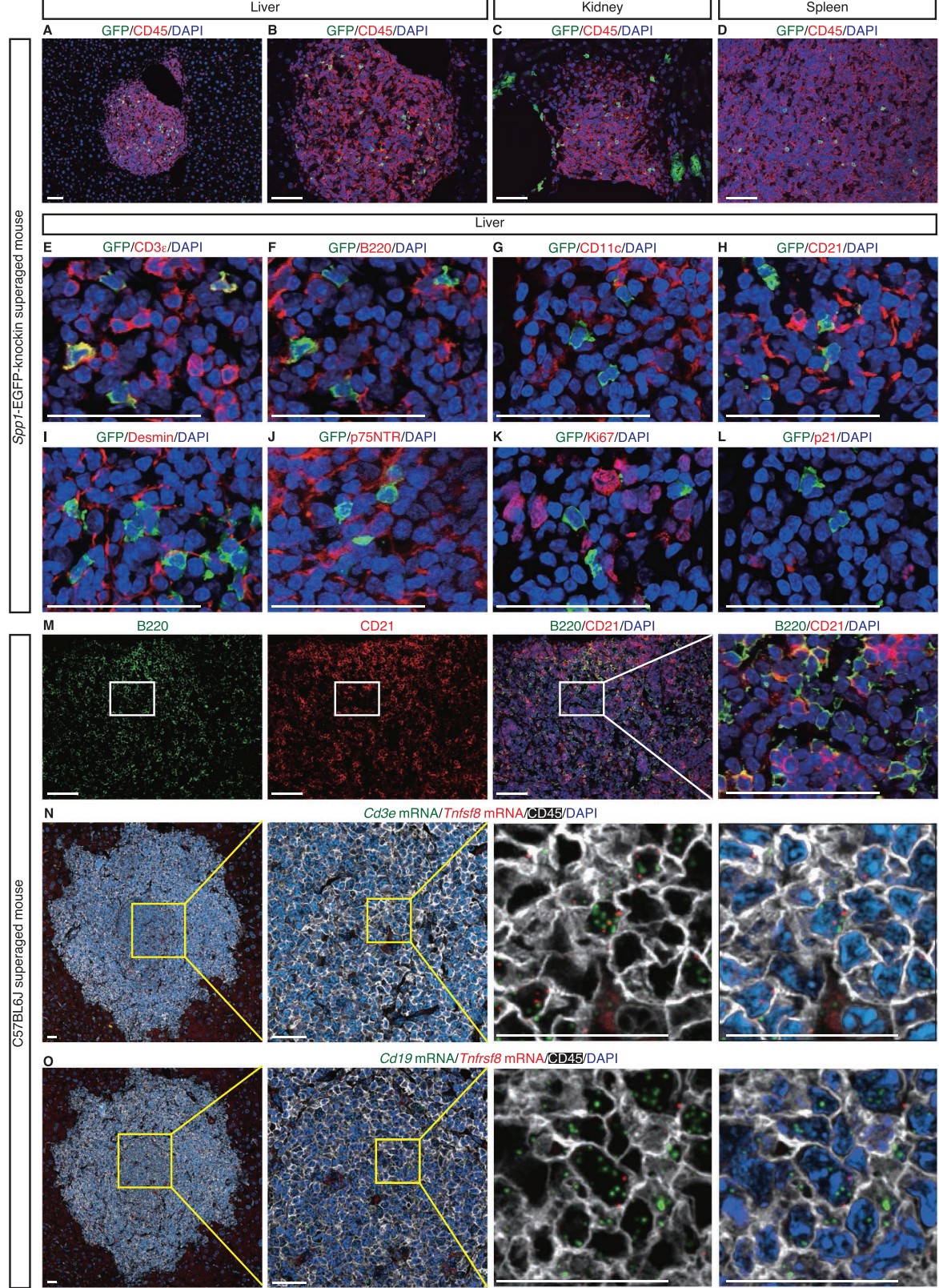

**Fig 2. SAT cells and ABCs expand within hepatic TLTs in superaged mice.** Immunofluorescence analysis of GFP and CD45 in the liver (A, B), kidney (C), and spleen (D) of superaged *Spp1*-EGFP-KI mice. Immunofluorescence analysis of (E) GFP and CD3ε; (F) GFP

and B220; (G) GFP and CD11c (a dendritic cell marker); (H) GFP and CD21; (I) GFP and Desmin; (J) GFP and p75NTR; (K) GFP and Ki67; and (L) GFP and p21 in the livers of superaged *Spp1*-EGFP-KI mice, and (M) B220 and CD21 in the livers of superaged C57BL6J mice. *In situ* hybridization images of (N) *Cd3e* and *Tnfsf8* with immunofluorescence co-staining for CD45; and (O) *Cd19* and *Tnfrsf8* with immunofluorescence co-staining for CD45 in the livers of superaged C57BL6J mice. Scale bars in (A-M): 50 μm; in (N-O): 25 μm.

most GFP$^+$ cells were negative for Ki67 or p21 (Fig 2K, L). These characteristics were similar to those of SAT cells in the kidney [12]. *Spp1* was also expressed in the bile ducts in the liver (S2 Fig). Moreover, we detected CD21$^-$ B cells, which are thought to include ABCs, within hepatic TLTs, as described in kidney TLTs (Fig 2M) [12]. Additionally, *in situ* hybridization showed that SAT cells (CD45$^+$*Tnfsf8*$^+$*Cd3e*$^+$ cells) and ABCs (CD45$^+$*Tnfrsf8*$^+$*Cd19*$^+$ cells) were present within TLTs in the liver (Fig 2N, O) [9]. Taken together, we confirmed the localization of SAT cells and ABCs within hepatic TLTs.

## TLT-related gene expression is upregulated in HCV patients and is associated with liver injury and fibrosis

Next, we examined whether TLT-related genes were expressed in human livers using the publicly available RNA-seq dataset [38]. Previously we have shown a strong correlation between the expression of *Tnf*, *Infg*, *Cxcl13*, *Ccl19*, and TLT sizes in the kidneys [9]. The expression levels of TLT-related genes, such as *CD3E*, *MS4A1* (*CD20*), *TNF*, *INFG*, *CXCL13*, and *CCL19* were upregulated in patients with HCV infection compared with patients without HCV infection, suggesting the possibility that more hepatic TLTs were formed in patients with HCV (Fig 3A). Expression levels of *TNFSF8* (*CD153*), a marker of SAT cells, and *TNFRSF8* (*CD30*), a marker of ABCs, were also upregulated in patients with HCV infection, although the upregulation of *TNFRSF8* was not statistically significant. On the other hand, the expression of these genes was not upregulated in patients with HBV infection compared with patients without HBV infection (Fig 3B) [39].

Furthermore, the expression of *CXCL13*, previously shown to have a strong and positive correlation with TLT size in the kidney [9] and found to have a moderate and positive correlation with TLT sizes in the liver in this study (Fig 1S), was positively correlated with the expression of *TNFSF8* and *TNFRSF8*, respectively (Fig 3C). Additionally, the expression of *TNFSF8* and *TNFRSF8* was also positively correlated (Fig 3C). These data also implicate the possible involvement of CD153–CD30 signaling in hepatic TLT formation in humans.

Interestingly, the expression levels of *CXCL13*, *TNFSF8*, and *TNFRSF8* correlated negatively with the expression of *retinol binding protein 4* (*RBP4*) (Fig 3D), which is decreased in cirrhosis [40], and positively with the expression of *COL1A1*, which reflects liver fibrosis (Fig 3E). These findings suggest that hepatic TLT formation could be correlated with liver injury and fibrosis.

## Hepatic TLTs in patients with HCV infection are associated with aging

To further investigate whether hepatic TLTs were frequently observed in the livers of patients with HCV infection, and to analyze the correlation between TLT formation and tissue injury, we performed histological analysis of the livers of patients with HCV infection. We found several inflammatory cell aggregates near the portal tracts and central veins (Fig 4A,B). We also observed severe fibrosis around these inflammatory cell aggregates (Fig 4C), which is consistent with the previous report that profibrotic fibroblast resided outside TLTs [41]. Immunofluorescence analysis showed that these inflammatory cell aggregates were composed mainly of T cells and B cells (Fig 4D), among which Ki67$^+$ signals were also detected (Fig 4E). In addition, some of these inflammatory cell aggregates contained CD21$^+$ and CD23$^+$ FDC

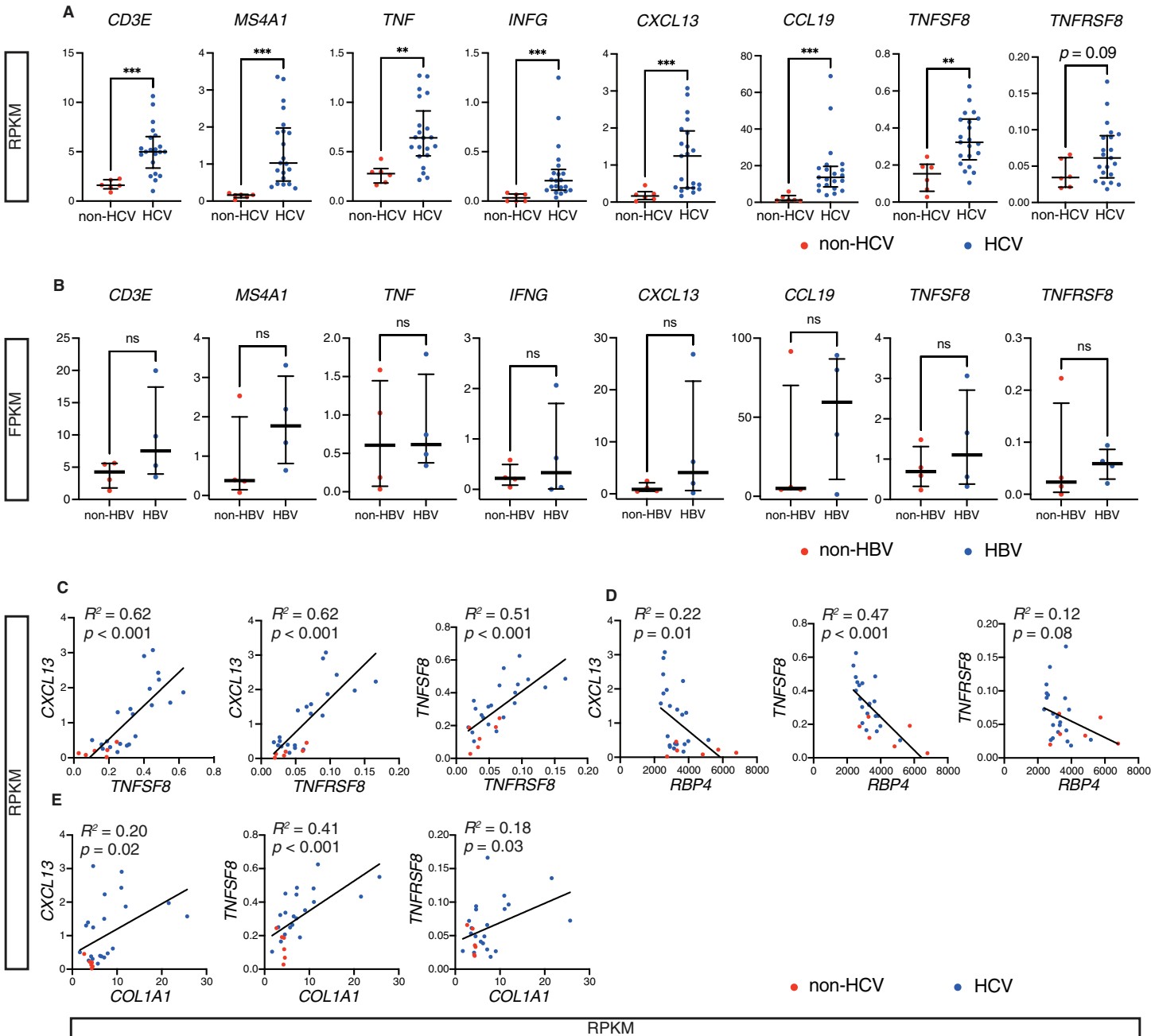

**Fig 3. TLT-related gene expression is upregulated in the liver of HCV patients and is associated with liver injury and fibrosis.** (A, C-E) Reanalysis of published RNA-seq data of human liver samples of patients with HCV infection (HCV, n = 6) or without HCV infection (non-HCV, n = 21) and (B) reanalysis of published RNA-seq data of human samples of patients with HBV infection (HBV, n = 4) or without HBV infection (non-HBV, n = 4). In (A, B) data are shown as the median and interquartile range. Mann–Whitney test is used to analyze the difference (A, B). Correlations in (C–E) are determined by Pearson's correlation analysis. **$p \leq 0.01$, ***$p \leq 0.001$. Abbreviation: RPKM, reads per kilobase of exon per million mapped reads; FPKM, fragments per kilobase per million mapped fragments.

networks (Fig 4E, F). These results suggest that some inflammatory cell aggregates function as TLTs. Similar to our findings in mice, TLTs contained Desmin⁺ fibroblasts (Fig 4G) and arteries (Fig 4H). CXCL13⁺ signals were also detected in non-hematopoietic cells within TLTs (Fig 4I). *In situ* hybridization further confirmed the signals for *TNFSF8* (CD153, a marker of

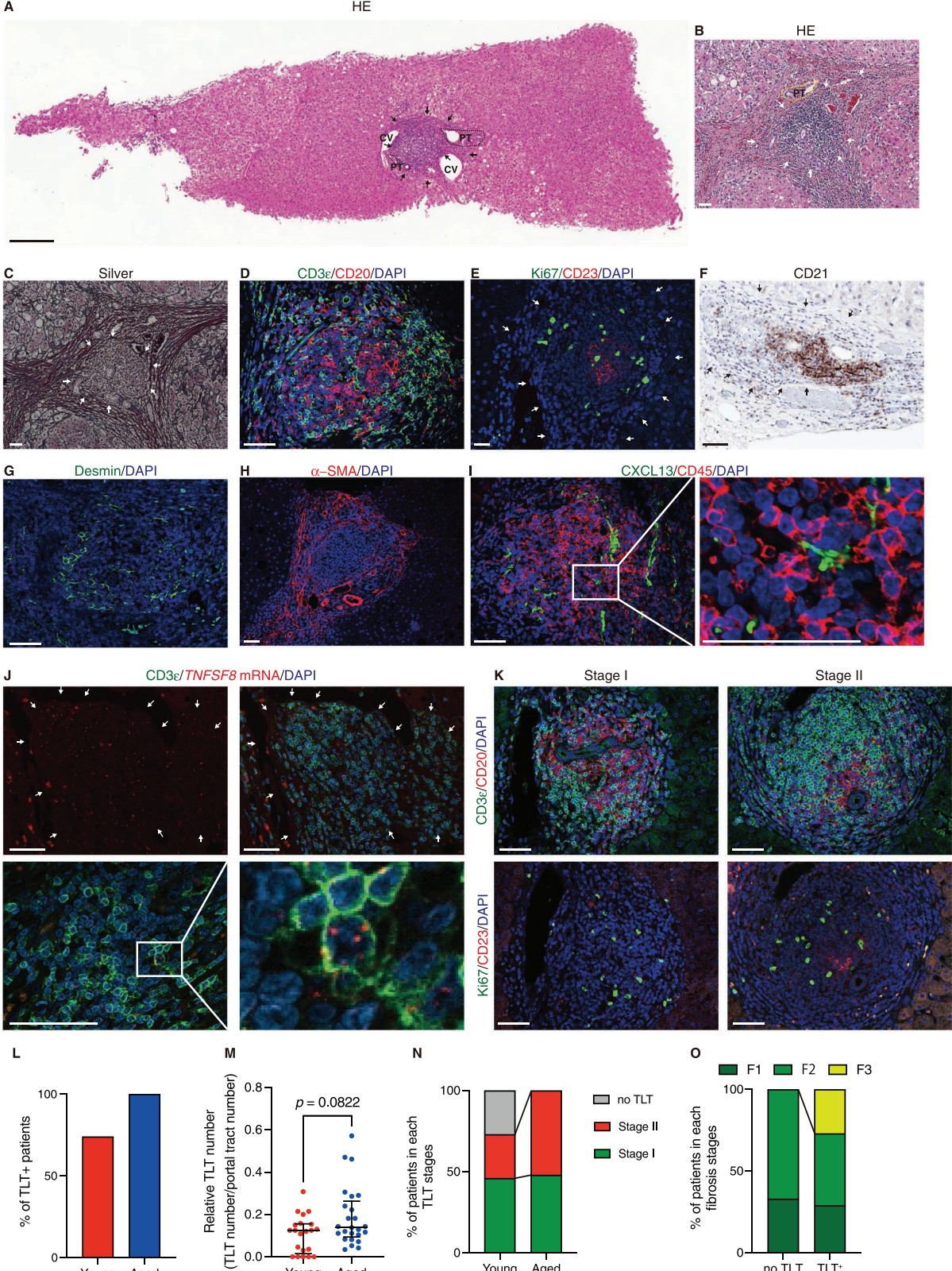

**Fig 4. Hepatic TLTs in HCV patients.** (A–K) Histological analyses of HCV patients' liver samples. (A, B) Hematoxylin and eosin (H&E) and (C) silver stain. Immunofluorescence analysis of (D) CD3ε (a T-cell marker) and CD20 (a B-cell marker); (E) Ki67 and CD23 (a

follicular dendritic cell marker); (G) Desmin (a satellite cell marker); (H) α-SMA and (I) CXCL13 (a homeostatic chemokine) and CD45. (F) Immunohistochemical analysis of CD21 (a follicular dendritic cell marker). (J) *In situ* hybridization images of *TNFSF8* with immuno-fluorescence co-staining for CD3ε (K) Representative immunofluorescence analysis of two distinct stages (I and II) of TLTs. Stage I TLTs are the aggregation of lymphocytes with the sign of proliferation, and do not include FDCs or germinal centers; Stage II TLTs include FDCs, but not germinal center. Immunofluorescence analysis of CD3ε and CD20; Ki67 and CD23 in each stage are serial sections. (L–N) Quantification of hepatic TLTs in young and aged patients with HCV infection (n = 22 and 25, respectively). (L) TLT frequencies in young and aged patients are 72.7% and 100%, respectively. (M) The relative numbers of TLTs per portal tract number. The medians of those in young and aged patients are 0.125 and 0.138, respectively. (N) The proportion of patients in each TLT stages. The proportion of patients with stage I TLTs in young and aged patients are 45% and 48%, respectively, while that of patients with stage II TLTs are 27% and 52%, respectively. (O) Percentage of patients in each fibrosis stage according to the new Inuyama classification stratified by hepatic TLT presence (no TLT: n = 6, with TLTs: n = 41). Arrows indicate TLT localization. Scale bars: in (A) 200 μm; (B–K) 50 μm. Abbreviations: CV, central vein; PT, portal tract.

SAT cells) within TLTs, some of which colocalized with CD3ε signals (Fig 4J). We also classified TLTs as in mice, and found stages I and II, but not stage III, TLTs in our cohort (Fig 4K). These results suggest that cellular and molecular components of hepatic TLTs in patients with HCV infection were similar to those in mouse livers. Furthermore, hepatic TLTs in patients with HCV infection exhibited various stages of developmental maturation.

To analyze the effect of hepatic TLTs on liver function and outcome, we examined liver biopsies samples prior to interferon therapy from 47 patients with HCV infection. Table 1 presents baseline patients' characteristics stratified by hepatic TLT presence and stages. None of them were diagnosed with hepatocellular carcinoma. Among a total of 47 patients, we detected hepatic TLTs in 41 patients (87.2%), of whom 22 were classified as stage I and 19 were stage II (Table 1). The median age of patients in each group was 53 years for patients without TLTs, 62 years for patients with stage I TLTs, and 63 years for patients with stage II TLTs, respectively (Table 1). While none of the patients without TLTs have been previously treated with interferon, some of the patients with TLTs have been previously treated with interferon. In addition, ALBI scores were −3.03 for patients without TLTs, −2.90 for patients with stage I TLTs and −2.73 for patients with stage II TLTs. Furthermore, FIB-4 index was 1.59 for patients without TLTs, 2.61 for patients with stage I TLTs and 2.55 for patients with stage II TLTs.

Overall, 100% of the aged patients (60 years of age or over) were positive for hepatic TLTs, while 72.7% of the young patients were positive for hepatic TLTs (Fig 4L). The number of TLTs in the liver biopsy samples was not statistically different between two groups (Fig 4M). The percentage of patients with stage II TLTs was 52% in aged patients and 27% in young patients (Fig 4N).

Next, we investigated the relationship between hepatic TLTs and the new Inuyama classification (Table 2), which classifies the activity grades based on the degree of lymphocytes infiltration and hepatocyte necrosis from A0 (no necro-inflammatory reaction) to A3 (severe necro-inflammatory reaction), and the fibrosis stages from F0 (no fibrosis) to F4 (cirrhosis) [35]. The percentage of patients with severe fibrosis (F3) was 26.8% in patients with hepatic TLTs and 0% in patients without hepatic TLTs (Fig 4O). All patients had no difference in activity grade (Table 2).

We also investigated the relationship between hepatic TLTs and outcomes of 47 HCV patients. Among them, 25 patients were treated with interferon, and the ratio of patients who achieved negative for serum HCV RNA at 24 weeks after the end of interferon therapy (SVR24) was 73% for patients with stage I TLTs and 78% for patients with stage II TLT, compared with 100% for patients without TLTs (S3 Fig A). Among 47 HCV patients, we were able to follow the prognosis of 32 patients. Within five years after liver biopsy, TLT formation was not associated with the cumulative incidence of hepatocellular carcinoma (S3 Fig B).

**Table 1. Clinical characteristics of patients evaluated in the study.**

| | Overall (N = 47) | no TLT (N = 6) | TLT stage I (N = 22) | TLT stage II (N = 19) | *P*-value |
|---|---|---|---|---|---|
| Age, yr (IQR) | 60 (49, 66) | 53 (43, 55) | 62 (49, 66) | 63 (54, 67) | 0.093 |
| Aged patients (%) | 25 (53.2) | 0 (0) | 12 (54.5) | 13 (68.4) | NE[d] |
| Gender, male (%) | 21 (44.7) | 4 (66.7) | 9 (40.9) | 8 (42.1) | |
| Previous interferon therapy, n (%) | 8 (17.0) | 0 (0) | 5 (22.7) | 3 (15.8) | NE[d] |
| PLT, ×10³/μL (IQR) | 16.9 (12.4, 21.4) | 18.0 (14.8, 21.9) | 15.5 (11.6, 21.6) | 17.2 (12.9, 22.9) | 0.568 |
| GOT, IU/L (IQR) | 48 (36,71) | 60 (27, 75) | 43 (36, 67) | 48 (36, 110) | 0.953 |
| GPT, IU/L (IQR) | 56 (36, 100) | 93 (27, 121) | 53 (32, 97) | 55 (43, 130) | 0.769 |
| ALP, IU/L (IQR) | 280 (218, 334) | 308 (267, 337) | 278 (211, 339) | 272 (204, 345) | 0.671 |
| γGTP, IU/L (IQR) | 44 (22, 96) | 36 (18, 127) | 67 (22, 86) | 38 (24, 109) | 0.755 |
| ALB[a], g/dL (IQR) | 4.3 (4.1, 4.6) | 4.6 (4.4, 4.8) | 4.3 (3.8, 4.6) | 4.3 (4.0, 4.6) | 0.167 |
| T-Bil, mg/dL (IQR) | 0.6 (0.5, 0.8) | 0.6 (0.5, 0.8) | 0.6 (0.5, 0.7) | 0.7 (0.5, 0.8) | 0.509 |
| ALBI score[a] (IQR) | −2.84 (−3.00, −2.58) | −3.03 (−3.23, −2.82) | −2.90 (−3.12, −2.44) | −2.73 (−2.91, −2.59) | 0.177 |
| ALBI grade[a] | | | | | |
| Grade 1, n (%) | 33/46 (71.7) | 5/6 (83.3) | 14/21 (66.7) | 14/19 (73.7) | 0.739 |
| Grade 2, n (%) | 13/46 (28.3) | 1/6 (16.7) | 7/21 (33.3) | 5/19 (26.3) | |
| FIB-4 index (IQR) | 2.41 (1.54, 3.92) | 1.59 (1.21, 2.38) | 2.61 (1.59, 4.19) | 2.55 (1.32, 3.92) | 0.216 |
| HCV RNA[b] (IQR) | 6.3 (5.6, 6.6) | 6.6 (6.1, 6.8) | 5.9 (4.1, 6.6) | 6.5 (5.7, 6.7) | 0.120 |
| HCV serotype[c] | | | | | |
| I, n (%) | 22/37 (59.5) | 3/5 (60.0) | 9/16 (56.3) | 10/16 (62.5) | 1.000 |
| II, n (%) | 15/37 (40.5) | 2/5 (40.0) | 7/16 (43.8) | 6/16 (37.5) | |

[a]There is one missing value for ALB, ALBI score, and ALBI grade in TLT stage I group.

[b]There is one missing value for HCV RNA in no TLT group.

[c]There are one, six, and three missing values for HCV serotype in no TLT group, TLT stage I group, and TLT stage II group, respectively.

[d]Because of the existence of a zero cell count, we did not performed Fisher's exact test.

Abbreviations: ALB, albumin; ALBI, albumin–bilirubin; ALP, alkaline phosphatase; FIB-4, fibrosis-4; GOT, glutamic oxaloacetic transaminase; GPT, glutamic pyruvic transaminase; GTP, glutamyl transpeptidase; PLT, platelet; T-Bil, total-bilirubin; TLT, tertiary lymphoid tissue; yr, year old, NE, not evaluated.

## Discussion

In this study, we showed that cellular and molecular components of hepatic TLTs were similar to those of renal TLTs, and that SAT cells and ABCs were also present in hepatic TLTs. Furthermore, analysis of publicly available data on the transcriptomes of human liver biopsies revealed that the expression of TLT-related genes was elevated in liver biopsies from patients with HCV and was associated with liver injury and fibrosis.

The cellular and molecular components of hepatic TLTs were similar to those of renal TLTs. Furthermore, the presence of SAT cells and ABCs and the correlation between the expression of their surface markers and TLT-related genes suggest that these unique lymphocytes may also contribute to hepatic TLT formation. The localization of hepatic TLTs was in the vicinity of central veins and portal veins, similar to the perivascular formation of renal TLTs. These results suggest that the mechanism of TLT formation is common across tissues.

It has been known that lymphoid follicles/aggregates occur in 47–78% of liver biopsies from patients with HCV [24], but their definition was unclear and varied among reports, and

**Table 2. Histological characteristics of patients evaluated in the study.**

| | Overall (N = 47) | no TLT (N = 6) | TLT stage I (N = 22) | TLT stage II (N = 19) |
|---|---|---|---|---|
| Number of portal tracts, n (IQR) | 24 (15, 31) | 19 (15, 32) | 23 (14, 30) | 27 (14, 36) |
| New Inuyama criteria | | | | |
| Activity grade | | | | |
| A0, n (%) | 0 (0) | 0 (0) | 0 (0) | 0 (0) |
| A1, n (%) | 17 (36.2) | 2 (33.3) | 9 (40.9) | 6 (31.6) |
| A2, n (%) | 30 (63.8) | 4 (66.7) | 13 (59.1) | 13 (68.4) |
| A3, n (%) | 0 (0) | 0 (0) | 0 (0) | 0 (0) |
| Fibrosis stage | | | | |
| F0, n (%) | 0 (0) | 0 (0) | 0 (0) | 0 (0) |
| F1, n (%) | 14 (29.8) | 2 (33.3) | 6 (27.3) | 6 (31.6) |
| F2, n (%) | 22 (46.8) | 4 (66.7) | 11 (50.0) | 7 (36.8) |
| F3, n (%) | 11 (23.4) | 0 (0) | 5 (22.7) | 6 (31.6) |
| F4, n (%) | 0 (0) | 0 (0) | 0 (0) | 0 (0) |
| TLT number per portal tract (IQR) | 0.13 (0.05, 0.20) | 0 (0, 0) | 0.11 (0.05, 0.14) | 0.16 (0.14, 0.24) |
| Stage I TLT number per portal tracts (IQR) | 0.10 (0.04, 0.14) | 0 (0, 0) | 0.11 (0.05, 0.14) | 0.11 (0.07, 0.14) |
| Stage II TLT number per portal tracts (IQR) | 0.00 (0.00, 0.05) | 0 (0, 0) | 0 (0, 0) | 0.07 (0.04, 0.11) |

the association with liver prognosis has been controversial. With a clear definition of TLTs, we found that hepatic TLTs were observed more frequently than previously considered, especially in aged patients, in whom TLTs were present in 100% of patients. This suggests that aging, as in the kidney, plays an important role in the formation of hepatic TLTs in patients with HCV. Analysis from publicly available datasets also confirmed that the expression of TLT-related genes was associated with liver injury and fibrosis. In addition, 26.8% in patients with hepatic TLTs had severe fibrosis, whereas 0% in patients without hepatic TLTs had severe fibrosis. Due to the presence of a zero cell, statistical testing could not be performed. These results are consistent with our finding that renal TLTs are associated with aging, kidney injury and poor prognosis [13,14].

Some previous reports have not shown a positive correlation between the presence of lymphoid follicles/aggregates and fibrosis [23,25 ]. This discrepancy may be because the definition of lymphoid follicles in previous reports was not based on the molecular definition or quantitative analysis, and therefore may include simple inflammatory cell infiltration. In addition, these reports have not further classified lymphoid follicles according to their maturational stages. From another perspective, Kumagai *et al.* evaluated only portal fibrosis, whereas we analyzed fibrosis according to the new Inuyama classification [23].

Lymphoid aggregate, some of which are TLTs, can be found in patients not only with HCV but also with hepatitis B virus infection, autoimmune hepatitis, and primary biliary cirrhosis [42–44]. Furthermore, lymphoid aggregates are known to be more common in hepatitis C than in hepatitis B [26,45–47 ] or autoimmune hepatitis [48]. Boldanova *et al.* compared gene expression changes between patients with and without HCV infection and found that biological processes such as lymphocyte and leukocyte activation were upregulated in patients with

HCV infection [38]. Based on the gene expression changes in publicly available datasets in this study, TLTs are more likely to be formed in patients with HCV.

Our study has several limitations. First, this human cohort was a small single-center study, which limits the generalizability of our findings. Because of the small number of cases, we could not clarify the relationship between hepatic TLTs and the prognosis of liver function. In recent years, patients with HCV have tended not to undergo a liver biopsy, which is one reason for this small number of cases. Second, in our human cohort, non-HCV patients were not included, however, as for autoimmune liver diseases, it is speculated that TLTs might maintain and propagate autoimmunity by providing an environment where self-antigens can be presented, and antigen-specific T cells can be activated [43]. Analysis of the role of hepatic TLTs in autoimmune hepatitis may contribute to an enhanced understanding of the relationship between liver disease and hepatic TLT. Finally, there was a possibility of underestimation of TLTs with only two serial sections of liver biopsies. Nevertheless, we obtained several insights regarding the similarities of cellular and molecular components of liver TLTs with those of renal TLTs and the association with aging.

Our findings demonstrate that the cellular and molecular components of hepatic TLTs are similar to those of renal TLTs and suggest that hepatic TLTs might be associated with aging, liver injury and liver fibrosis. Although the frequency of liver biopsies in HCV patients is expected to decrease further in the future, we have identified serum and urine biomarkers of TLTs (manuscript in preparation), which will allow us to further explore their relationship with prognosis and other aspects of the disease in the future.

## Supporting information

**S1 Table. Primer sequences used for real-time PCR.**
(DOCX)

**S2 Table. Data sets of mice analyzed in this study.**
(XLSX)

**S1 Fig. Portal fibroblasts are not common within hepatic TLTs.** Immunofluorescence of CD90.2 (a portal fibroblast and a T-cell marker) and CD45 around bile ducts (A, B) and within TLTs (C, D). Abbreviation: BD, bile duct. Scale bars; 50 μm.
(TIF)

**S2 Fig. Spp1 is expressed in part of the bile ducts.** Immunofluorescence of GFP and cytokeratin (CK) 19 (a bile duct marker) in the liver of superaged Spp1-EGFP-Knockin mice. EGFP is visualized with an anti-GFP antibody. Scale bar: 50 μm.
(TIF)

**S3 Fig. Interferon therapy response and cumulative incidence of hepatocellular carcinoma in patients with or without TLTs.** (A) The proportion of patients achieving sustained virological response at week 24 (SVR24) after interferon treatment in patients without TLTs, with Stage I TLTs and with Stage II TLTs (n = 5, 11 and 9, respectively). The proportion of patients with SVR24 in patients without TLTs, with Stage I TLTs and with Stage II TLTs are 100%, 73% and 78%, respectively. (B) Cumulative incidence of hepatocellular carcinoma (HCC) in patients with or without TLTs after liver biopsy (n = 27 and 5, respectively). The cumulative incidences of HCC in patients with TLTs are 4.17, 4.17, and 8.73% at 1, 3, and 5 years after the liver biopsy, respectively. Those without TLTs are 0, 0, and 0% at 1, 3, and 5 years after the liver biopsy, respectively. Log-rank test is used to compare the cumulative incidence of HCC.
(TIF)

## Acknowledgments

We thank Ms. Yamashita, Ms. Ishizaki, Ms. Sakurai, Ms. Okamoto, Ms. Nakayama, Ms. Setsuda, and Ms. Ozone for their excellent technical assistance.

## Author contributions

**Conceptualization:** Naoya Toriu, Yuki Sato, Hiroteru Kamimura, Motoko Yanagita.

**Data curation:** Naoya Toriu, Shinya Yamamoto, Motoko Yanagita.

**Formal analysis:** Naoya Toriu, Shingo Fukuma.

**Funding acquisition:** Motoko Yanagita.

**Investigation:** Naoya Toriu, Yuki Sato, Hiroteru Kamimura, Takahisa Yoshikawa, Masaou Tanaka.

**Project administration:** Motoko Yanagita.

**Resources:** Hiroteru Kamimura, Masakazu Hattori, Shuji Terai.

**Supervision:** Motoko Yanagita.

**Writing – original draft:** Naoya Toriu, Yuki Sato.

**Writing – review & editing:** Naoya Toriu, Yuki Sato, Hiroteru Kamimura, Takahisa Yoshikawa, Masaou Tanaka, Shinya Yamamoto, Shingo Fukuma, Masakazu Hattori, Shuji Terai, Motoko Yanagita.

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
