## [Decision Letter · Decision Letter 0]

12 Jul 2024

PONE-D-24-18840Aligning Cellular and Molecular Components in Age-Dependent Tertiary Lymphoid Tissues of Kidney and LiverPLOS ONE

Dear Dr. Yanagita,

Thank you for submitting your manuscript to PLOS ONE. After careful consideration, we feel that it has merit but does not fully meet PLOS ONE’s publication criteria as it currently stands. Therefore, we invite you to submit a revised version of the manuscript that addresses the points raised during the review process.

**ACADEMIC EDITOR: **

Dear Authors,

please revise proposed manuscript thoroughly according to all reviewers' comments.

Additionally, please do the following:

- Visualization of obtained results must be improved

- Motivation behind proposed research should be more clearly explain. Please elaborate what is "beyond state-of-the-art" of proposed. study.

- Make sure that the source code is available according to PLOS ONE publication policies.

- Make sure that you have conducted rigid statistical analysis.

All the best,

Nebojsa Bacanin

We look forward to receiving your revised manuscript.

Kind regards,

Nebojsa Bacanin

Academic Editor

PLOS ONE

 [This research was supported by the Japan Agency for Medical Research and Development (AMED) under Grant Number AMED-CREST 19gm0610011, 22gm1210009, 22zf0127003h001, 22lm0203006, and 22ek0310020h0001; KAKENHI Grant-in-Aids for Scientific Research B (20H03697). This work was partly supported by the World Premier International Research Center Initiative (WPI), MEXT, Japan.].  

4. In the online submission form, you indicated that [The datasets of mice used and analyzed in the current study are available from the corresponding author upon reasonable request. The human datasets that support the findings of this study are available from the corresponding author, but restrictions apply to the availability of these data, which were used under license for the current study, and so are not publicly available. Data are however available from the authors upon reasonable request and with permission of the ethics committee at Niigata University and the ethical committee at Kyoto University Hospital. Previously published RNA-seq data are available in GEO (GSE84346: https://www.ncbi.nlm.nih.gov/geo/query/acc.cgi?acc=GSE84346 and GSE171294: https://www.ncbi.nlm.nih.gov/geo/query/acc.cgi?acc=GSE171294).]. 

5. We note that Figures 1, 2, and 4 in your submission contain [map/satellite] images which may be copyrighted. All PLOS content is published under the Creative Commons Attribution License (CC BY 4.0), which means that the manuscript, images, and Supporting Information files will be freely available online, and any third party is permitted to access, download, copy, distribute, and use these materials in any way, even commercially, with proper attribution. For these reasons, we cannot publish previously copyrighted maps or satellite images created using proprietary data, such as Google software (Google Maps, Street View, and Earth). For more information, see our copyright guidelines: http://journals.plos.org/plosone/s/licenses-and-copyright.

1. You may seek permission from the original copyright holder of Figures 1, 2, and 4 to publish the content specifically under the CC BY 4.0 license.  

Additional Editor Comments:

Dear Authors,

please revise proposed manuscript thoroughly according to all reviewers' comments.

Additionally, please do the following:

- Visualization of obtained results must be improved

- Motivation behind proposed research should be more clearly explain. Please elaborate what is "beyond state-of-the-art" of proposed. study.

- Make sure that the source code is available according to PLOS ONE publication policies.

- Make sure that you have conducted rigid statistical analysis.

All the best,

Nebojsa Bacanin

Reviewers' comments:

Reviewer's Responses to Questions

**Comments to the Author**

1. Is the manuscript technically sound, and do the data support the conclusions?

Reviewer #1: Yes

Reviewer #2: Partly

2. Has the statistical analysis been performed appropriately and rigorously? 

Reviewer #1: Yes

Reviewer #2: No

3. Have the authors made all data underlying the findings in their manuscript fully available?

Reviewer #1: Yes

Reviewer #2: Yes

4. Is the manuscript presented in an intelligible fashion and written in standard English?

Reviewer #1: Yes

Reviewer #2: Yes

5. Review Comments to the Author

Reviewer #1: The authors have demonstrated that TLTs developed in the liver with aging, and that their cellular and molecular components were similar to those in the kidneys, such as senescence-associated T cells and age-associated B cells. Additionally, the authors have shown that the expression of TLT-related genes was elevated in the liver biopsy samples from HCV-infected patients compared with those without HCV infection. TLT formation was also associated with liver injury and fibrosis.

The present study suggests that TLT formation is observed across the tissues over aging. Although the causal relationship between TLT formation, chronic inflammation and aging remains unclear, further understanding of this cross-organ phenomenon may lead to solutions to many diseases and health problems.

HCV-related chronic hepatitis is now actively treated and curable disease. Is TLT formation associated with treatment response or liver function outcome?

Is TLT formation associated with other liver dysfunction or disease such as non-C virus hepatitis and alcoholic/non-alcoholic/drug-induced/primary biliary cholangitis?

P18 LL389 “liver fibrosis x” � “liver fibrosis”

Reviewer #2: The present manuscript is about tertiary lymphoid tissues (TLT) in the liver. The authors found aging and HCV were associated with TLT formation in the liver, and might contribute to liver dysfunction and fibrosis. The topic is interesting, but some critical concerns should be addressed.

1. Which type of non-hematopoietic cells express CXCL13 in TLT in the liver tissue?

2. The interpretation of results and conclusion are too strong because statistical analyses have not been performed in Tables 1 and 2. Also, it seems to be difficult to describe TLT increases and develop to more advanced stages based on the data in Figure 4L-O. I understand that they are due to small sample size, but this point is critical because we should discuss based on data.

6. PLOS authors have the option to publish the peer review history of their article (what does this mean? ). If published, this will include your full peer review and any attached files.

**Do you want your identity to be public for this peer review?** For information about this choice, including consent withdrawal, please see our Privacy Policy .

Reviewer #1: No

Reviewer #2: **Yes: ** Satoshi Hara

---

## [Author Response · Author response to Decision Letter 1]

31 Aug 2024

Point by point reply to the Editor and reviewers’ comments and suggestions

We would like to thank the editors and the reviewers for their valuable and insightful comments and the useful suggestions that have helped us to improve our manuscript.

- Visualization of obtained results must be improved

Reply to the academic editor:

Thank you for your important comments. We paid attention to the visualization quality of newly added data (Fig. 1R-S, S3_Fig)

- Motivation behind proposed research should be more clearly explain. Please elaborate what is "beyond state-of-the-art" of proposed study.

Reply to the academic editor: I appreciate your important comment. We have previously clarified the cellular and molecular components and clinical significance of TLTs in the kidney, and are curious whether our findings on renal TLTs could also be shared across the tissue. In this study, we therefore analyzed the cellular and molecular components of TLTs in the liver and their clinical significance, and found that there are similarities with TLTs in the kidney. We add the following sentence in the revised manuscript:

“Although TLTs develop with aging in other organs as well, their cellular and molecular components, and clinical significance remain unclear.” (p.2, l.7-8)

“we tried to characterize the cellular and molecular components of hepatic TLTs using superaged mouse and human livers, and to elucidate the clinical relevance of them.” (p.4, l.2-3)

“demonstrate that the cellular and molecular components of hepatic TLTs are similar to those of renal TLTs and” (p.20, l.18-19)

- Make sure that the source code is available according to PLOS ONE publication policies.

Reply to the academic editor: Thank you for your suggestion. We have no source code for sharing.

- Make sure that you have conducted rigid statistical analysis.

Reply to the academic editor: Thank you for your suggestion. We performed statistical analysis in Table 1. In Table 2, Fig4 L, N-0, we could not perform statistical analysis because of the existence of a zero cell count.

Reply to the academic editor: Thank you for your suggestion. We followed the PLOS ONE’s style requirements.

Reply to the academic editor: Thank you for your comment. We added the following sentence in the “Material and methods” section in the revised manuscript:

“Informed consent was obtained using an opt-out process on the website. Minors were not included among the patients.” (p.8, l.11-12)

[This research was supported by the Japan Agency for Medical Research and Development (AMED) under Grant Number AMED-CREST 19gm0610011, 22gm1210009, 22zf0127003h001, 22lm0203006, and 22ek0310020h0001; KAKENHI Grant-in-Aids for Scientific Research B (20H03697). This work was partly supported by the World Premier International Research Center Initiative (WPI), MEXT, Japan.].

Reply to the academic editor: Thank you for your suggestion. We added the following statement in the revised manuscript and cover letter.

"The funders had no role in study design, data collection and analysis, decision to publish, or preparation of the manuscript." (p.21, l.10-11)

4. In the online submission form, you indicated that [The datasets of mice used and analyzed in the current study are available from the corresponding author upon reasonable request. The human datasets that support the findings of this study are available from the corresponding author, but restrictions apply to the availability of these data, which were used under license for the current study, and so are not publicly available. Data are however available from the authors upon reasonable request and with permission of the ethics committee at Niigata University and the ethical committee at Kyoto University Hospital. Previously published RNA-seq data are available in GEO (GSE84346: https://www.ncbi.nlm.nih.gov/geo/query/acc.cgi?acc=GSE84346 and GSE171294: https://www.ncbi.nlm.nih.gov/geo/query/acc.cgi?acc=GSE171294).].

Reply to the academic editor: Thank you for your suggestion. We included the mouse datasets in the supplement. However, as for human dataset, the ethics committees at Niigata University and Kyoto University Hospital have not approved uploading the human datasets because public availability would compromise patient privacy. Still, data are available from the authors upon reasonable request and with permission of the ethics committees at Niigata University and Kyoto University Hospital.

We changed the data availability statement as follows:

“The mouse datasets are available in the Supporting Information files (S2_Table).” (p.22, l.9)

“The institutional contacts are: ethics@adm.niigata-u.ac.jp (the ethics committee at Niigata University) and ethcom@kuhp.kyoto-u.ac.jp (the ethical committee at Kyoto University Hospital).” (p.22, l.14-15)

5. We note that Figures 1, 2, and 4 in your submission contain [map/satellite] images which may be copyrighted.

Reply to the academic editor: We apologize for not being clear for copyright. All figures included in our paper were taken by the authors and are not copyrighted by another organization.

Reviewer: 1

Comments to the Author

The authors have demonstrated that TLTs developed in the liver with aging, and that their cellular and molecular components were similar to those in the kidneys, such as senescence-associated T cells and age-associated B cells. Additionally, the authors have shown that the expression of TLT-related genes was elevated in the liver biopsy samples from HCV-infected patients compared with those without HCV infection. TLT formation was also associated with liver injury and fibrosis.

The present study suggests that TLT formation is observed across the tissues over aging. Although the causal relationship between TLT formation, chronic inflammation and aging remains unclear, further understanding of this cross-organ phenomenon may lead to solutions to many diseases and health problems.

HCV-related chronic hepatitis is now actively treated and curable disease. Is TLT formation associated with treatment response or liver function outcome?

Reply to the reviewer 1: Thank you for your valuable and supportive comments and important questions. Unfortunately, this study did not collect the data on the usage of direct acting antivirals (DAAs). Combination therapy with DAA and interferon has been available in Japan since 2011. Among the patients analyzed, 37 patients received liver biopsy before 2011, and the remaining 10 patients received liver biopsy after 2011. Of these 10 patients, four patients on treated with interferon therapy, but the use of DAA is unknown. On the other hand, there are data about the treatment response to interferon therapy (newly added S3_Fig). The sustained virologic response at 24 weeks (SVR24) after interferon therapy was 100% for patients without TLTs, 73% for patients with stage I TLTs, and 78% for patients with stage II TLTs (S3_Fig A). While we could not perform Fisher’s exact test due to zero cell counts, there might be a tendency for TLTs in the liver to be associated with poor response to interferon therapy (S3_Fig A). On the other hand, TLT formation was not associated with the cumulative incidence of hepatocellular carcinoma (S3_Fig B).

We add the following sentences and S3_Fig in the revised manuscript.

“Kaplan-Meier method and log-rank test were used to depict and compare the cumulative incidence of hepatocellular carcinoma (HCC).” (p. 9, l.22-23)

“We also investigated the relationship between hepatic TLTs and outcomes of 47 HCV patients. Among them, 25 patients were treated with interferon, and the ratio of patients who achieved negative for serum HCV RNA at 24 weeks after the end of interferon therapy (SVR24) was 73% for patients with stage I TLTs and 78% for patients with stage II TLT, compared with 100% for patients without TLTs (S3_Fig A). Among 47 HCV patients, we were able to follow the prognosis of 32 patients. Within five years after liver biopsy, TLT formation was not associated with the cumulative incidence of hepatocellular carcinoma (S3_Fig B).” (p. 17, l.8-14)

Is TLT formation associated with other liver dysfunction or disease such as non-C virus hepatitis and alcoholic/non-alcoholic/drug-induced/primary biliary cholangitis?

Reply to the reviewer 1: Thank you for your comment. As you suggested, TLT formation is associated with other liver disease such as HBV hepatitis (Link A et al. Am J Pathol. 2011;178(4):1662-75) autoimmune hepatitis, and primary biliary cholangitis (Pikarsky E et al. Gastroenterology. 2016;151(5):780-3, van Buuren N et al. JHEP Rep. 2022;4(1):100388).

We added the following sentence in the revised manuscript:

“Lymphoid aggregate, some of which are TLTs, can be found in patients not only with HCV but also with hepatitis B virus infection, autoimmune hepatitis, and primary biliary cirrhosis [41-43].” (p. 20, l.2-3)

P18 LL389 “liver fibrosis x” � “liver fibrosis”

Reply to the reviewer 1: Thank you for your kind comment and we are sorry for the typo. We corrected the sentence.

Reviewer #2:

Comments to the Author

The present manuscript is about tertiary lymphoid tissues (TLT) in the liver. The authors found aging and HCV were associated with TLT formation in the liver, and might contribute to liver dysfunction and fibrosis. The topic is interesting, but some critical concerns should be addressed.

1. Which type of non-hematopoietic cells express CXCL13 in TLT in the liver tissue?

Reply to the reviewer 2: Thank you for your important comment. Immunostaining analysis showed the colocalization of CXCL13 and Desmin and CD21, respectively (Fig. 1R, S), which suggest that Desmin+ fibroblasts and CD21+ follicular dendritic cells (FDCs) express CXCL13. We added the following sentence.

“CXCL13 was expressed in Desmin+ fibroblasts and CD21+ FDCs (Fig 1R, S).” (p. 10, l.17-18)

2. The interpretation of results and conclusion are too strong because statistical analyses have not been performed in Tables 1 and 2. Also, it seems to be difficult to describe TLT increases and develop to more advanced stages based on the data in Figure 4L-O. I understand that they are due to small sample size, but this point is critical because we should discuss based on data.

Reply to the reviewer 2: Thank you for your important suggestion, and we apologize for the too-strong interpretation. We performed statistical analysis in Table 1. Unfortunately, in Table 2, Figure 4L, and Figure 4N-O, we could not perform Fisher’s exact test due to the existence of zero cell counts. Furthermore, we modified the result and discussion sections to avoid strong statement on the data without statistical significance.

We changed the result and discussion as follows.

“The median age of patients in each group was 53 years for patients without TLTs, 62 years for patients with stage I TLTs, and 63 years for patients with stage II TLTs, respectively (Table 1). While none of the patients without TLTs have been previously treated with interferon, some of the patients with TLTs have been previously treated with interferon. In addition, ALBI scores were −3.03 for patients without TLTs, −2.90 for patients with stage I TLTs and −2.73 for patients with stage II TLTs. Furthermore, FIB-4 index was 1.59 for patients without TLTs, 2.61 for patients with stage I TLTs and 2.55 for patients with stage II TLTs.” (p. 15, l.15-21)

“Overall, 100% of the aged patients (60 years of age or over) were positive for hepatic TLTs, while 72.7% of the young patients were positive for hepatic TLTs (Fig. 4L). The number of TLTs in the liver biopsy samples was not statistically different between two groups (Fig. 4M). The percentage of patients with stage II TLTs was 52% in aged patients and 27% in young patients (Fig. 4N).” (p. 16, l.12- p. 17 l.1)

“The percentage of patients with severe fibrosis (F3) was 26.8% in patients with hepatic TLTs and 0% in patients without hepatic TLTs (Fig. 4O). All patients had no difference in activity grade (Table 2).” (p. 17, l.5-7)

“With a clear definition of TLTs, we found that hepatic TLTs were observed more frequently than previously considered, especially in aged patients, in whom TLTs were present in 100% of patients.” (p. 19, l.15-17)

“Our findings demonstrate that the cellular and molecular components of hepatic TLTs are similar to those of renal TLTs and suggest that hepatic TLTs might be associated with aging, liver injury and liver fibrosis.” (p. 20, l.18-19)

Although the association between TLTs and liver fibrosis was not statistically supported in Figure 4, we reserved the sentence because of the association in Figure 3. To soften our statement, we changed "are" to "might be".

---

## [Decision Letter · Decision Letter 1]

16 Dec 2024

PONE-D-24-18840R1Aligning Cellular and Molecular Components in Age-Dependent Tertiary Lymphoid Tissues of Kidney and LiverPLOS ONE

Dear Dr. Yanagita,

Thank you for submitting your manuscript to PLOS ONE. After careful consideration, we feel that it has merit but does not fully meet PLOS ONE’s publication criteria as it currently stands. Therefore, we invite you to submit a revised version of the manuscript that addresses the points raised during the review process.

We look forward to receiving your revised manuscript.

Kind regards,

Elsayed Seddek Ibrahem Mohammed, Ph.D.

Academic Editor

PLOS ONE

Journal Requirements:

Additional Editor Comments:

Dear Authors,

On behalf of PLOS ONE, I would like to extend my gratitude for submitting your manuscript entitled “Aligning Cellular and Molecular Components in Age-Dependent Tertiary Lymphoid Tissues of Kidney and Liver” (Manuscript ID: PONE-D-24-18840R1). Your work represents a significant contribution to the understanding of the dynamic interplay between cellular and molecular components within tertiary lymphoid tissues and their age-related adaptations in the kidney and liver. I understand the limitations that you have faced in your study and the effort that you have put to get sufficient and informative data.

After a thorough review process, I am pleased to inform you that your manuscript has been accepted for publication, pending the incorporation of minor corrections. These revisions will enhance the clarity and impact of your findings. Please see the attached document outlining the necessary modifications.

We greatly appreciate your efforts and commend the scientific rigor and innovation reflected in your work. If you have any questions or require assistance during the revision process, please do not hesitate to reach out.

Thank you once again for contributing your research to PLOS ONE. We look forward to the final version of your manuscript and its forthcoming publication.

Editor comments to the authors:

1- Page 5 line 96 : the Formaline used in fixation was 4% paraformaldehyde or 4% NBF,, Please describe

2- Page 5 line 97: the model and manufacturer of the microtome used in this study.

3- Page 5 line 97: the reference of the stain H and E and staining protocol (for example Bancroft et al., or Savana et al., 2018)

4- Page 5 line 103:the cryostat model and manufacturer is missing.

5- Page 6 line 117: the used formaldehyde for fixation is it the same in all the experiments / if so Please unify in all the manuscript as mentioned in the first comment.

Reviewer comments to the author:

Dr. Yanagita and colleagues submitted their manuscript, "Aligning Cellular and Molecular Components in Age-Dependent Tertiary Lymphoid Tissues of Kidney and Liver," for review. The study addresses a significant topic in immunology and aging, offering novel insights into the systemic formation of tertiary lymphoid tissues (TLTs). The manuscript is scientifically sound and presents an important advancement in the understanding of TLTs in aging and liver pathology.

Recommendations are provided to refine and enhance the manuscript for publication.

1. Figures and Visual Representation

(1) Figure 1:

• Use uniform scale bars across sub-panels, positioned consistently (e.g., bottom-right).

• Clarify differences between TLT stages (I, II, III) in panel (V) with added markers or legends.

(2) Figure 4:

• Improve legends for panels (L–N) to explain the relationship between age and TLT stages.

(3) Supplementary Figures:

• Ensure all axes in the figures (e.g., S3) are clearly labeled, with legends comprehensively describing the content.

2. Interpretation of Results

• While the manuscript successfully tempers conclusions due to small sample sizes in some analyses, consider further nuanced discussions of statistical limitations. For instance: Clarify the significance of TLT presence in fibrosis-related outcomes when statistical significance is not achieved (e.g., Figure 4).

3. Clinical Applications and Generalizability

• Strengthen the discussion on the broader implications of TLT findings in conditions beyond HCV, such as non-alcoholic steatohepatitis or autoimmune hepatitis.

Once these minor issues are addressed, I am confident that the manuscript will meet the high standards of PLOS ONE and greatly benefit its readership.

Thank you for the opportunity to review this excellent work. Please do not hesitate to contact me if further clarification is needed.

Best Regard's,

Reviewers' comments:

Reviewer's Responses to Questions

**Comments to the Author**

1. If the authors have adequately addressed your comments raised in a previous round of review and you feel that this manuscript is now acceptable for publication, you may indicate that here to bypass the “Comments to the Author” section, enter your conflict of interest statement in the “Confidential to Editor” section, and submit your "Accept" recommendation.

Reviewer #1: All comments have been addressed

Reviewer #2: All comments have been addressed

Reviewer #3: All comments have been addressed

2. Is the manuscript technically sound, and do the data support the conclusions?

Reviewer #1: Yes

Reviewer #2: Yes

Reviewer #3: Yes

3. Has the statistical analysis been performed appropriately and rigorously? 

Reviewer #1: Yes

Reviewer #2: Yes

Reviewer #3: Yes

4. Have the authors made all data underlying the findings in their manuscript fully available?

Reviewer #1: Yes

Reviewer #2: Yes

Reviewer #3: No

5. Is the manuscript presented in an intelligible fashion and written in standard English?

Reviewer #1: Yes

Reviewer #2: Yes

Reviewer #3: Yes

6. Review Comments to the Author

Reviewer #1: (No Response)

Reviewer #2: (No Response)

Reviewer #3: Drs. Yanagita and colleagues submitted their manuscript, "Aligning Cellular and Molecular Components in Age-Dependent Tertiary Lymphoid Tissues of Kidney and Liver," for review. The study addresses a significant topic in immunology and aging, offering novel insights into the systemic formation of tertiary lymphoid tissues (TLTs). The manuscript is scientifically sound and presents an important advancement in the understanding of TLTs in aging and liver pathology.

Recommendations are provided to refine and enhance the manuscript for publication.

1. Figures and Visual Representation

(1) Figure 1:

• Use uniform scale bars across sub-panels, positioned consistently (e.g., bottom-right).

• Clarify differences between TLT stages (I, II, III) in panel (V) with added markers or legends.

(2) Figure 4:

• Improve legends for panels (L–N) to explain the relationship between age and TLT stages.

(3) Supplementary Figures:

• Ensure all axes in the figures (e.g., S3) are clearly labeled, with legends comprehensively describing the content.

2. Interpretation of Results

• While the manuscript successfully tempers conclusions due to small sample sizes in some analyses, consider further nuanced discussions of statistical limitations. For instance: Clarify the significance of TLT presence in fibrosis-related outcomes when statistical significance is not achieved (e.g., Figure 4).

3. Clinical Applications and Generalizability

• Strengthen the discussion on the broader implications of TLT findings in conditions beyond HCV, such as non-alcoholic steatohepatitis or autoimmune hepatitis.

7. PLOS authors have the option to publish the peer review history of their article (what does this mean? ). If published, this will include your full peer review and any attached files.

**Do you want your identity to be public for this peer review?** For information about this choice, including consent withdrawal, please see our Privacy Policy .

Reviewer #1: No

Reviewer #2: **Yes: ** Satoshi Hara

Reviewer #3: No

---

## [Author Response · Author response to Decision Letter 2]

17 Jan 2025

Point by point reply to the Editor and reviewers’ comments and suggestions

We would like to thank the editors and the reviewers for their valuable and insightful comments and the useful suggestions that have helped us to improve our manuscript.

1- Page 5 line 96 : the Formaline used in fixation was 4% paraformaldehyde or 4% NBF,, Please describe

Reply to the editor: Thank you for your comment.

We used 10% NBF and described the fixation method in the Methods.

2- Page 5 line 97: the model and manufacturer of the microtome used in this study.

Reply to the editor: Thank you for your comment.

We described the model and manufacturer of the microtome.

3- Page 5 line 97: the reference of the stain H and E and staining protocol (for example Bancroft et al., or Savana et al., 2018)

Reply to the editor: Thank you for your comment.

We cited the reference for the staining protocol.

4- Page 5 line 103:the cryostat model and manufacturer is missing.

Reply to the editor: Thank you for your comment.

We described the model and manufacturer of the cryostat.

5- Page 6 line 117: the used formaldehyde for fixation is it the same in all the experiments / if so Please unify in all the manuscript as mentioned in the first comment.

Reply to the editor: Thank you for your comments.

The used formaldehyde for fixation is the same in all the experiments. We described the fixation method.

1. Figures and Visual Representation

(1) Figure 1:

• Use uniform scale bars across sub-panels, positioned consistently (e.g., bottom-right).

Reply to the reviewer 3: Thank you for your suggestion.

We unified the scale bars, and positioned at the bottom-left consistently.

• Clarify differences between TLT stages (I, II, III) in panel (V) with added markers or legends.

Reply to the reviewer 3: Thank you for your valuable comments.

We added the following sentences to the figure legends.

“Stage I TLTs are the aggregation of lymphocytes with the sign of proliferation, and do not include FDCs or germinal centers; Stage II TLTs include FDCs, but not germinal centers; Stage III TLTs include FDCs and germinal centers.” (p.11, l.13-15)

“Stage I TLTs are the aggregation of lymphocytes with the sign of proliferation, and do not include FDCs or germinal centers; Stage II TLTs include FDCs, but not germinal centers.” (p.14, l.26-28)

(2) Figure 4:

• Improve legends for panels (L–N) to explain the relationship between age and TLT stages.

Reply to the reviewer 3: Thank you for your valuable comments.

We changed the legend for panels (L-N) as follows:

“(L) TLT frequencies in young and aged patients are 72.7% and 100%, respectively. (M) The relative numbers of TLTs per portal tract number. The medians of those in young and aged patients are 0.125 and 0.138, respectively. (N) The proportion of patients in each TLT stages. The proportion of patients with stage I TLTs in young and aged patients are 45% and 48%, respectively, while that of patients with stage II TLTs are 27% and 52%, respectively.” (p.15, l.2-6)

(3) Supplementary Figures:

• Ensure all axes in the figures (e.g., S3) are clearly labeled, with legends comprehensively describing the content.

Reply to the reviewer 3: Thank you for your valuable comments.

We changed the axis of S3 Fig A to “% of patients with SVR24 (%)”. We also corrected the typo of the y axis of S3 Fig B. In addition, we added the sentence in the legend.

“The proportion of patients with SVR24 in patients without TLTs, with Stage I TLTs and with Stage II TLTs are 100%, 73% and 78%, respectively.” (p.27, l.33-34)

“The cumulative incidences of HCC in patients with TLTs are 4.17, 4.17, and 8.73% at 1, 3, and 5 years after the liver biopsy, respectively. Those without TLTs are 0, 0, and 0% at 1, 3, and 5 years after the liver biopsy, respectively. Log-rank test is used to compare the cumulative incidence of HCC.” (p.27, l.36-39)

Furthermore, we changed the label of Fig1W and S1 Fig as follows.

“% of each stage among all TLTs” (Fig 1W)

“CD90.2 CD45 DAPI” (S1 Fig)

2. Interpretation of Results

• While the manuscript successfully tempers conclusions due to small sample sizes in some analyses, consider further nuanced discussions of statistical limitations. For instance: Clarify the significance of TLT presence in fibrosis-related outcomes when statistical significance is not achieved (e.g., Figure 4).

Reply to the reviewer 3: Thank you for your valuable and supportive comments

We added the following sentence.

In addition, 26.8% in patients with hepatic TLTs had severe fibrosis, whereas 0% in patients without hepatic TLTs had severe fibrosis. Due to the presence of a zero cell, statistical testing could not be performed.” (p.19, l.20-22)

3. Clinical Applications and Generalizability

• Strengthen the discussion on the broader implications of TLT findings in conditions beyond HCV, such as non-alcoholic steatohepatitis or autoimmune hepatitis.

Reply to the reviewer 3: Thank you for your valuable and supportive comments.

It is reported that non-alcoholic steatohepatitis is not associated with hepatic TLTs in human patients and mouse models (Pikarsky E, et al. Gastroenterology. 2016;151(5):780-3.). Therefore, we discussed about autoimmune hepatitis.

We added the following sentence.

“however, as for autoimmune liver diseases, it is speculated that TLTs might maintain and propagate autoimmunity by providing an environment where self-antigens can be presented, and antigen-specific T cells can be activated [43]. Analysis of the role of hepatic TLTs in autoimmune hepatitis may contribute to an enhanced understanding of the relationship between liver disease and hepatic TLT.” (p.20, l.15-19)

---

## [Decision Letter · Decision Letter 2]

30 Jan 2025

Aligning Cellular and Molecular Components in Age-Dependent Tertiary Lymphoid Tissues of Kidney and Liver

PONE-D-24-18840R2

Dear Dr. Yanagita,

We’re pleased to inform you that your manuscript has been judged scientifically suitable for publication and will be formally accepted for publication once it meets all outstanding technical requirements.

Kind regards,

Elsayed Seddek Ibrahem Mohammed, Ph.D.

Academic Editor

PLOS ONE

Additional Editor Comments (optional):

Subject: Acceptance of Manuscript PONE-D-24-18840R2

Dear editor ,

We are pleased to inform you that your manuscript titled "Aligning Cellular and Molecular Components in Age-Dependent Tertiary Lymphoid Tissues of Kidney and Liver" (Manuscript ID: PONE-D-24-18840R2) has been accepted for publication in PLOS ONE.

Your study provides a significant contribution to our understanding of the cellular and molecular dynamics of tertiary lymphoid tissues, especially in the context of age-dependent changes. The thorough analysis and presentation of findings in this manuscript exemplify the high standard of research we strive to publish.

Before finalizing the publication, we would like to request a minor revision to Reference 31:

Current:

Kim SS, Layton C, Bancroft JD (2018) Bancroft’s theory and practice of histological techniques, 8th edn. Elsevier, Amsterdam.

Revised:

Suvarna KS, Layton C, Bancroft JD (2018). Bancroft's theory and practice of histological techniques. Elsevier health sciences.

Please ensure this revision is incorporated in your reference list before publication. Additionally, kindly review the references thoroughly to ensure accuracy and compliance with the journal’s formatting guidelines.

Congratulations on your successful submission! We appreciate your contribution to the field and look forward to showcasing your work to the broader scientific community.

Sincerely,

Elsayed Mohammed

PLOS ONE Editorial Team

Reviewers' comments:

Reviewer's Responses to Questions

**Comments to the Author**

1. If the authors have adequately addressed your comments raised in a previous round of review and you feel that this manuscript is now acceptable for publication, you may indicate that here to bypass the “Comments to the Author” section, enter your conflict of interest statement in the “Confidential to Editor” section, and submit your "Accept" recommendation.

Reviewer #3: All comments have been addressed

2. Is the manuscript technically sound, and do the data support the conclusions?

Reviewer #3: Yes

3. Has the statistical analysis been performed appropriately and rigorously? 

Reviewer #3: Yes

4. Have the authors made all data underlying the findings in their manuscript fully available?

Reviewer #3: Yes

5. Is the manuscript presented in an intelligible fashion and written in standard English?

Reviewer #3: Yes

6. Review Comments to the Author

Reviewer #3: Dr. Yanagita and his team revised and resubmitted the manuscript, 'Aligning Cellular and Molecular Components in Age-Dependent Tertiary Lymphoid Tissues of Kidney and Liver.' I have no further concerns about this paper and recommend it for publication.

7. PLOS authors have the option to publish the peer review history of their article (what does this mean? ). If published, this will include your full peer review and any attached files.

**Do you want your identity to be public for this peer review?** For information about this choice, including consent withdrawal, please see our Privacy Policy .

Reviewer #3: No

---

## [Editor Report · Acceptance letter]

PONE-D-24-18840R2

PLOS ONE

Dear Dr. Yanagita,

I'm pleased to inform you that your manuscript has been deemed suitable for publication in PLOS ONE. Congratulations! Your manuscript is now being handed over to our production team.

Kind regards,

on behalf of

Dr. Elsayed Seddek Ibrahem Mohammed

Academic Editor

PLOS ONE